# Anion-Dominated Copper Salicyaldimine Complexes—Structures, Coordination Mode of Nitrate and Decolorization Properties toward Acid Orange 7 Dye

**DOI:** 10.3390/polym12091910

**Published:** 2020-08-24

**Authors:** Meng-Jung Tsai, Chi-Jou Tsai, Ken Lin, Jing-Yun Wu

**Affiliations:** Department of Applied Chemistry, National Chi Nan University, Nantou 545, Taiwan; s97324905@mail1.ncnu.edu.tw (M.-J.T.); tgb751@hotmail.com (C.-J.T.); rivernate1009@gmail.com (K.L.)

**Keywords:** acid orange 7, anion effect, copper, nitrate, degradation

## Abstract

A salicyaldimine ligand, 3-tert-butyl-4-hydroxy-5-(((pyridin-2-ylmethyl)imino)methyl)benzoic acid (H_2_L_salpyca_) and two Cu(II)−salicylaldimine complexes, [Cu(HL_salpyca_)Cl] (**1**) and [Cu(HL_salpyca_)(NO_3_)]*_n_* (**2**), have been synthesized. Complex **1** has a discrete mononuclear structure, in which the Cu(II) center is in a distorted square-planar geometry made up of one HL_salpyca_^−^ monoanion in an NNO tris-chelating mode and one Cl^−^ anion. Complex **2** adopts a neutral one-dimensional zigzag chain structure propagating along the crystallographic [010] direction, where the Cu(II) center suits a distorted square pyramidal geometry with a *τ* value of 0.134, consisted of one HL_salpyca_^−^ monoanion as an NNO tris-chelator and two NO_3_^−^ anions. When the Cu∙∙∙O semi coordination is taken into consideration, the nitrato ligand bridges two Cu(II) centers in an unsymmetrical bridging-tridentate with a *μ*, *κ*^4^O,O′:O′,O″ coordination. Clearly, anion herein plays a critical role in dominating the formation of discrete and polymeric structures of copper salicyaldimine complexes. Noteworthy, complex **2** is insoluble but highly stable in H_2_O and various organic solvents (CH_3_OH, CH_3_CN, acetone, CH_2_Cl_2_ and THF). Moreover, complex **2** shows good photocatalytic degradation activity and recyclability to accelerate the decolorization rate and enhance the decolorization performance of acid orange 7 (AO7) dye by hydrogen peroxide (H_2_O_2_) under daylight.

## 1. Introduction

Salicyaldimine derivatives with the NO donor set of azomethine nitrogen and phenolato oxygen have been extensively studied in coordination chemistry due to their preparational accessibilities and their versatility in terms of steric and electronic modifications [1,2,3,4,5]. Metal–salicyaldimine complexes so far have achieved a remarkable success in structure variation including traditional coordination complexes [2,3,4,6,7] and metallosupramolecular architectures such as discrete macrocycles [8,9], helicates [10,11] and coordination polymers [12,13,14,15]. Moreover, metal–salicyaldimine complexes have wide applications in fields such as catalysis [5,15,16], molecular magnetism [7,17] and fluorescent chemosensors [18,19]. 

On the other hand, the presence of organic pollutants, such as azo dyes, the most diverse group of synthetic dyes used in food, textile and printing industries, in water streams is becoming a seriously environmental problem due to the fact of their color, toxicity, potential carcinogenicity and non-biodegradable nature [20,21,22,23,24]. Therefore, the treatment of dye-containing wastewater is a matter of great important issue for environmental protection and has attracted worldwide attention. Nowadays, adsorptive removal (non-destructive process) [25,26] and degradation (destructive process) [27,28] are the two most promising techniques widely utilized in the decolorization of dye-contaminated water owing to their advantages of mild operation conditions, convenience, high efficiency and relative low costs [20,22,23,24,29,30,31]. 

In recent years, our group has reported several transition metal complexes of salicyaldimine Schiff base ligands with NO and N_2_O donor sets, which have represented mononuclear molecular structures and one- and two-dimensional periodical structures (1D chain and 2D layer) with interesting structural and photophysical properties [14,32,33]. Further, our group has also contributed efforts in the decolorization of aqueous solutions of organic dyes through the adsorptive removal process [34,35,36,37]. As a continuation, we report herein the synthesis and structures of two new Cu(II)−salicylaldimine complexes, [Cu(HL_salpyca_)Cl] (**1**, H_2_L_salpyca_ = 3-tert-butyl-4-hydroxy-5-(((pyridin-2-ylmethyl)imino)methyl)benzoic acid) and [Cu(HL_salpyca_)(NO_3_)]*_n_* (**2**). Complex **1** shows a mononuclear structure while complex **2** has a 1D zigzag chain structure. Such structural diversities are tentatively ascribed to the influence of anion. It is also noted that the nitrato in **2** suits a bridging-tridentate ligand with a coordination mode of *μ*, *κ*^4^O, O′:O′,O″ to bridge two Cu(II) ions to form the extended structure. Moreover, complex **2** is very stable in H_2_O and would cooperate with H_2_O_2_ to show remarkable photocatalytic activity toward degradation of acid orange 7 (AO7) under daylight. 

## 2. Experimental Section

### 2.1. Materials and Methods 

Chemical reagents were of reagent grade quality obtained from commercial sources (Alfa Aesar, Heysham, UK; ACROS, Pittsburgh, PA, USA; SHOWA, Saitama, Japan) and used as received without further purification. 5-*tert*-Butyl-3-formyl-4-hydroxybenzoic acid [38,39] were synthesized according to the reported literature. Proton nuclear magnetic resonance (^1^H NMR)^1^H NMR spectra were collected at room temperature by using a Bruker AMX-300 Solution-NMR spectrometer (Bruker, New Taipei, Taiwan) operating at 300 MHz. Chemical shifts are reported in parts per million (ppm) with reference to the residual protons of the deuterated solvent and coupling constants are given in hertz (Hz). Mass spectra were recorded by using a Bruker Daltonics flexAnalysis matrix-assisted laser desorption/ionization time of flight (MALDI-TOF) mass spectrometer (Bruker, New Taipei, Taiwan). Thermogravimetric (TG) analyses were carried out under a flux of nitrogen by using a Thermo Cahn VersaTherm HS TG analyzer (Thermo, Newington, NH, USA) at a heating rate of 5 °C per minute from 25 to 900 °C. X-ray powder diffraction (XRPD) patterns were acquired on a Shimadzu XRD-7000 diffractometer (Shimadzu, Kyoto, Japan) with a graphite monochromatized Cu Kα radiation (*λ* = 1.5406 Å; 40 kV, 30 mA) at a scan speed of 1.2° per minute. Infrared (IR) spectra were collected on a Perkin-Elmer Frontier Fourier transform infrared spectrometer (Perkin-Elmer, Taipei, Taiwan) in the 4000–500 cm^−1^ region using attenuated total reflection (ATR) technique. Microanalyses were carried out by using an Elementar Vario EL III analytical instrument (Elementar, Langenselbold, Germany). UV-vis absorption spectra were recorded on a JASCO V-750 UV/VIS spectrophotometer (JASCO, Tokyo, Japan). Fluorescence spectra were recorded at room temperature on a Hitachi F7000 fluorescence spectrophotometer (Hitachi, Tokyo, Japan). Inductively coupled plasma optical emission spectrometry (ICP-OES) analyses were conducted on an Agilent 5100 ICP-OES instrument (Agilent, Santa Clara, CA, US). 

### 2.2. Synthesis of 3-Tert-butyl-4-hydroxy-5-(((pyridin-2-ylmethyl)imino)methyl)benzoic Acid (H_2_L_salpyca_) 

To a methanolic solution (2 mL) of 2-(aminomethyl)pyridine (0.10 g, 1.0 mmol), a methanolic solution (6 mL) of 5-*tert*-butyl-3-formyl-4-hydroxybenzoic acid (0.22 g, 1.0 mmol) was added under nitrogen atmosphere. The solution was stirred for 2 h at room temperature. After the solvent was removed under reduced pressure, yellow powdered product of H_2_L_salpyca_ was obtained in a yield of 54% (0.17 g, 0.54 mmol). ^1^H NMR (300 MHz, DMSO-*d*_6_, ppm): *δ* 8.81 (s, 1H), 8.56 (d, *J* = 4.8 Hz, 1H), 7.96 (s, 1H), 7.86–7.80 (m, 2H), 7.44 (d, *J* = 7.8 Hz, 1H), 7.36–7.31 (m, 1H), 4.92 (s, 2H), 1.34 (s, 9H) (Appendix A). MS (MALDI-TOF): *m/z* 312.878 [M + H]^+^(calcd for C_18_H_20_N_2_O_3_: *m/z* 312.18) (Appendix A). IR (ATR, cm^−1^): 3425, 1961, 2924, 2862, 1956, 1677, 1632, 1602, 1477, 1392, 1364, 4334, 1307, 1279, 1239, 1202, 1179, 1120, 1014, 946, 879, 839, 804, 767, 705, 674, 602. Anal. Calcd for C_18_H_20_N_2_O_3_: C, 69.21; H, 6.45; N, 8.97%. Found: C, 69.14; H, 6.29; N, 8.68%. 

### 2.3. Synthesis of [Cu(HL_salpyca_)Cl] (***1***) 

A methanolic solution (1 mL) of H_2_L_salpyca_ (31.2 mg, 0. 10 mmol) and an aqueous solution (1 mL) of CuCl_2_ (13.4 mg, 0.10 mmol) were sealed in a Teflon-lined stainless steel container. The container was heated at 80 °C for 12 h and then cooled to 30 °C. Green needle-shaped crystals of **1** were obtained in a yield of 39% (16.3 mg, 3.9 × 10^−^^2^ mmol). MS (MALDI-TOF): *m/z* 410.806 [M]^+^(calcd for C_18_H_19_N_2_O_3_ClCu: *m/z* 410.34) (Appendix A). IR (ATR, cm^−1^): 3065, 2961, 2908, 2869, 2568, 1672, 1596, 1536, 1484, 1407, 1355, 1263, 1258, 1230, 1173, 1052, 996, 921, 824, 795, 761, 697, 658. TGA: 200 °C (decomp.). Anal. Calcd for C_18_H_19_N_2_O_3_ClCu: C, 52.68; H, 4.67; N, 6.83%. Found: C, 52.88; H, 4.39; N, 6.89%.

### 2.4. Synthesis of [Cu(HL_salpyca_)(NO_3_)]_n_ (***2***) 

A methanolic solution (0.5 mL) of H_2_L_salpyca_ (31.2 mg, 0. 10 mmol) and an aqueous solution (1.5 mL) of Cu(NO_3_)_2_∙2.5H_2_O (23.2 mg, 0.10 mmol) were sealed in a Teflon-lined stainless steel container. The container was heated at 80 °C for 12 h and then cooled to 30 °C. Green needle-shaped crystals of **2** were obtained in a yield of 28% (12.2 mg, 2.8 × 10^−^^2^ mmol). IR (ATR, cm^−1^): 3431, 2928, 2590, 2003, 1682, 1629, 1600, 1573, 1532, 1488, 1475, 1440, 1413, 1394, 1354, 1331, 1274, 1244, 1216, 1191, 1065, 1054, 1029, 1004, 948. TGA: 215 °C (decomp.). Anal. Calcd for C_18_H_19_N_3_O_6_Cu: C, 49.48; H, 4.38; N, 9.62%. Found: C, 49.45; H, 4.66; N, 9.47%. 

### 2.5. X-Ray Data Collection and Structure Refinement 

Quality crystals of **1** and **2** were obtained for single-crystal structure determination. Data collections were performed on an Oxford Diffraction Gemini S diffractometer for **1** and on a Bruker D8 Venture diffractometer for **2**. Both diffractometers are equipped with a graphite monochromated Mo K*α* radiation (*λ* = 0.71073 Å). Starting models for structure refinement were found using direct methods with SHELXS-97 program [40] and the structural data were refined by full-matrix least-squares methods against *F*^2^ using the SHELXL-2014/7 [41], incorporated in WINGX-v2014.1 [42] crystallographic collective package. All non-hydrogen atoms were found from the different Fourier maps and refined anisotropically. Carbon-bound hydrogen atoms were placed by geometrical calculation and refined as riding mode. Oxygen-bound hydrogen atoms were located in a difference Fourier map and the positional parameters were refined, with the restraint O–H = 0.82(1) Å. Isotropic displacement parameters of all hydrogen atoms were derived from the parent atoms. ORTEP plots and crystal structure drawings for **1** and **2** were drawn using the Diamond software [43]. CCDC 1955728–1955729 contain the supplementary crystallographic data for this paper. These data can be obtained free of charge via http://www.ccdc.cam.ac.uk/conts/retrieving.html or from the Cambridge Crystallographic Data Centre, 12 Union Road, Cambridge CB2 1EZ, UK; fax: (+44) 1223-336-033; or e-mail: deposit@ccdc.cam.ac.uk. Detailed crystallographic data are summarized in Table 1. Selected bond lengths and angles are listed in Table 2 and important hydrogen-bonding parameters are shown in Table 3. 

## 3. Results and Discussion

### 3.1. Syntheses and Characterization 

Complexes **1** and **2** were synthesized from the reactions of copper(II) salts (CuCl_2_ and Cu(NO_3_)_2_∙2.5H_2_O) and H_2_L_salpyca_ under hydro(solvo)thermal conditions at 80 °C for 12 h (Scheme 1). Their molecular structures have been determined by the single-crystal X-ray structure analyses and their bulky samples have been characterized to be a single crystalline phase through X-ray powder diffraction (XRPD) measurements (Appendix A). 

### 3.2. Crystal Structural Description of [Cu(HL_salpyca_)Cl] (***1***) 

Complex **1** crystallizes in the triclinic space group $P\bar{1}$ and the asymmetric unit consists of one Cu(II) center, one HL_salpyca_^−^ monoanion and one Cl^−^ anion. The Cu(II) center is tris-chelated by one HL_salpyca_^−^ monoanion through its phenolato, imino and pyridine groups and further coordinated by one Cl^−^ anion, furnishing a distorted square-planar geometry (Figure 1a), with the Cu and the Cl away from the plane 0.24 and 1.11 Å, respectively. The carboxyl group of the HL_salpyca_^−^ ligand is protonated and non-coordinated. The Cu–N_pyridine_, Cu–N_imino_, Cu–O_phenolato_ and Cu–Cl bond lengths are 1.9971(17), 1.9376(17), 1.8857(14) and 2.2299(6) Å, respectively and the N_pyridine_–Cu–N_imino_ and N_imino_–Cu–O_phenolato_ bond angles are 82.98(7) and 92.00(6)°, respectively. Two [Cu(HL_salpyca_)Cl] molecules form a hydrogen-bonded dimer through intermolecular hydrogen bonds (O3–H3∙∙∙O2#1, *d*(D∙∙∙A) = 2.637(2) Å, ∠(D–H∙∙∙A) = 168(3)°, #1 = −1−*x*, −*y*, −*z*) between two carboxyl groups of two HL_salpyca_^−^ ligands, with a graph set of $R_{2}^{2}\left(8\right)$ (Figure 1b) [44]. In addition, there are also intermolecular π∙∙∙π interactions (plane∙∙∙plane: 3.37 and 3.69 Å) between any two neighboring [Cu(HL_salpyca_)Cl] molecules that stack in a column manner along the crystallographic [010] direction, that is, *b* axis (Figure 1c). Moreover, there are also weak but significant intermolecular C–H∙∙∙Cl interactions (C9–H9B∙∙∙Cl2#2, *d*(D∙∙∙A) = 3.567(3) Å, ∠(D–H∙∙∙A) = 150°, #1 = −1 + *x*, *y*, *z*) between the chloro ligands and the methylene moieties of the HL_salpyca_^−^ ligands (Figure 1d). 

### 3.3. Crystal Structural Description of [Cu(HL_salpyca_)(NO_3_)]_n_ (***2***) 

Complex **2** crystallizes in the monoclinic space group *P*2_1_/*c* and has an infinite zigzag chain structure. The asymmetric unit consists of one Cu(II) center, one HL_salpyca_^−^ monoanion and one NO_3_^−^ anion. The Cu(II) center is coordinated by one HL_salpyca_^−^ monoanion as a tris-chelator through its phenolato, imino and pyridine groups and further coordinated to two NO_3_^−^ anion (Figure 2a), furnishing a distorted square pyramidal geometry, with a *τ* value of 0.134 [45]. The carboxyl group of the HL_salpyca_^−^ ligand is protonated and non-coordinated. The Cu–N_pyridine_, Cu–N_imino_, Cu–O_phenolato_ and Cu–O_nitrato_ bond lengths are 1.994(3), 1.931(2), 1.892(2) and 2.019(2)/2.437(3) Å, respectively and the N_pyridine_–Cu–N_imino_ and N_imino_–Cu–O_phenolato_ bond angles are 83.45(10) and 93.00(9)°, respectively. Basically, the nitrato ligand adopts a bridging-bidentate with a semi-coordination (*μ*, *κ*^2^O, O′). However, when the Cu∙∙∙O semi-coordination (Cu1∙∙∙O6 = 2.73 Å, Cu1∙∙∙O4#1 = 3.01, #1 = −*x*, *y* + 1/2, −*z* + 1/2) is taken into consideration, the nitrato ligand suits an unsymmetrical bridging-tridentate with a *μ*, *κ*^4^O, O′:O′,O″ coordination, with a Cu∙∙∙Cu separation of 4.97 Å, among various possible coordination modes of nitrato ligand [46,47,48,49,50,51]. Morozov and coworkers have shown that nitrato ligand exhibits 28 coordination modes that are classified in terms of the number of oxygen atoms used by the nitrato ligand for binding to complexing metal centers [52]. Based on the definition, M, B and T denote one, two and three nitrato O atoms used for binding, respectively and the superscripts mean the number of metal atoms connected to one nitrato O atom (the first superscript) or two nitrato O atoms (the second superscript). Accordingly, the coordination mode of nitrato ligand in complex **2** is classified to be T^02^ (Scheme 2, mode V). 

Each nitrato ligand bridges two [Cu(HL_salpyca_)]^+^cationic motifs to generate a neutral 1D zigzag chain structure that propagates along the crystallographic [010] direction, that is, the *b* axis (Figure 2b). There are chain-to-chain hydrogen bonds (O3–H3∙∙∙O2#2, *d*(D∙∙∙A) = 2.633(3) Å, ∠(D–H∙∙∙A) = 173°, #2 = 1 − *x*, 2 − *y*, −*z*) between two carboxyl groups of two HL_salpyca_^−^ ligands, with a graph set of $R_{2}^{2}\left(8\right)$ [44], in adjacent chains (Figure 2c). As a result, a hydrogen-bonded 2D wavy sheet is formed (Figure 2d). This structure is compared to a zigzag chain structure of the Cu(II)−salicylaldimine complex {[Cu(salpyca)]∙3H_2_O}*_n_* (where H_2_salpyca = 4-hydroxy-3-(((pyridin-2-yl)methylimino)methyl)benzoic acid) [33], prepared from a solvent-mediated cation-exchange of [Zn(salpyca)(H_2_O)]*_n_* and Cu(NO_3_)_2_∙6H_2_O in aqueous solution, where the salpyca^2−^ displays not only the NNO tris-chelating mode from its pyridine, imino and phenolato groups but also bridging behavior though its monodentately-carboxylato group. 

### 3.4. Thermogravimetric (TG) Analysis 

The thermogravimetric (TG) traces of **1** and **2** both showed a long plateau before a decomposition process occurred at temperature approaching 200 and 215 °C, respectively (Figure 3). For **2**, after a two-step decomposition of the framework ended at a temperature approaching 509 °C, a weight of 17.8% of the total sample was left which is reasonably assigned to CuO (calcd. 18.2%). 

### 3.5. Stability of ***2***


The chemical stability of zigzag chain **2** in crystalline phase has been examined. As observation, **2** displayed high chemical stability to maintain the framework integrity and crystallinity after immersing in H_2_O, methanol (CH_3_OH), acetonitrile (CH_3_CN), acetone, dichloromethane (CH_2_Cl_2_) and tetrahydrofuran (THF), respectively, for 1 day. This is supported by the checked XRPD profiles which are almost identical with that of as-synthesized crystalline samples (Figure 4). 

### 3.6. Photophysical Properties 

In solid state, the salicyaldimine ligand, H_2_L_salpyca_, shows an unstructured fluorescence centered at 502 nm (Appendix A), upon excitation at 360 nm, which is tentatively assigned to intraligand transition [32]. Comparably, the Cu(II) complexes **1** and **2** are fluorescence silent due to the paramagnetic perturbation and/or energy/charge transfer [2,53,54]. 

### 3.7. Decolorization of Acid Orange 7 (AO7) Dye 

Cu(II) complexes have shown the ability to various oxidation reactions including degradation of organic dyes [24,55,56,57,58,59]. Thus the decolorization properties of Cu(II) complexes **1** and 2 were evaluated via photocatalytic degradation of acid orange 7 (AO7) dye, a common azo dye, under natural conditions. One milligram of Cu(II) complex (**1** or 2) was dipped into to 3 mL of aqueous solutions of AO7 (20 mg L^−1^), which was magnetically stirring in the dark for 30 min. Then, an amount of 30% hydrogen peroxide (H_2_O_2_, 1 mL) was added under daylight. The UV/vis spectra of the reaction solution were measured at specific intervals with a UV/vis spectrophotometer and the AO7 concentrations were determined by the maximum absorbances at 485 nm. For comparison, the same experiments were also carried out in dark conditions. In addition, several AO7 decolorization experiments examined by bare (only AO7 in H_2_O without H_2_O_2_, **1** or **2**), H_2_O_2_-only, **1**-only and **2**-only were also conducted in dark conditions and under daylight. As observations, the absorbance of the major absorption band of AO7 in aqueous solutions remained almost constant in these checked decolorization experiments by bare, H_2_O_2_-only, **1**-only and **2**-only in dark conditions and under daylight (Appendix A). The results imply that AO7 cannot be adsorbed by **1** and **2** and is very difficultly photodecomposed by daylight irradiation or oxidative/photocatalytic degradation by H_2_O_2_ only or Cu(II) complex (**1** or 2) only in dark conditions and even under daylight irradiation. However, the degradation of AO7 was occurred in cases of simultaneous existence of H_2_O_2_ and Cu(II) complex (**1** or 2) in dark conditions and under daylight, as supported by time-dependent UV/vis spectra of AO7 after degradation (Figure 5a and Appendix A). In case of the simultaneous existence of **2** and H_2_O_2_ under daylight, the aqueous solutions of AO7 showed remarkable color change from orange at initial to very light after 60-min degradation and to colorless after about 480-min degradation under daylight (Figure 5b). The findings suggest an oxidative degradation process. Of particular note, the degradation rate and the degradation performance of AO7 by **1/**H_2_O_2_ and **2/**H_2_O_2_ are greatly improved and enhanced under daylight in compared to that in dark conditions, implying photocatalytic degradation of AO7 by **1/**H_2_O_2_ and **2/**H_2_O_2_. Moreover, the oxidative degradation ability of **2** is better than that of **1**. As a representative, the decolorization performances after 60-min degradation are about 3% and 10% for **1/**H_2_O_2_ and **2/**H_2_O_2_, respectively, in dark conditions and about 10% and 85% for **1/**H_2_O_2_ and **2/**H_2_O_2_, respectively, under daylight (Figure 5c). The decolorization performances would reach a maximum of approximately 15% for **1/**H_2_O_2_ and 25% for **2/**H_2_O_2_ in dark conditions and 25% for **1/**H_2_O_2_ and 98% for **2/**H_2_O_2_ under daylight. This is comparable with other cases of AO7 degradation reported in the literature [21,59,60,61,62,63,64,65]. 

To gain a better understanding of the degradation kinetics of AO7 catalyzed by **2/**H_2_O_2_, the experimental data were fitted by a first-order model as expressed by the following equation—ln (*C*/*C*_0_) = *kt*—where *C*_0_ and *C* are the initial and apparent concentration of AO7, respectively, *k* is the kinetic rate constant and *t* is time. As a result, the *k* values are determined to be 0.00291 min^−1^ and 0.0154 min^−1^ for degradation of AO7 in dark conditions and under daylight (Figure 6), respectively, within 30 min. Clearly, the rate constant of AO7 degradation under daylight is about five times larger than that in dark conditions, supporting again a photocatalytic degradation of AO7. Further, there exists the first-order exponential decay relationships between the concentration of AO7 (ppm) and degradation time (min), with the formulae of [AO7] = 3.97 × exp(−*t*/219.85) + 15.36 (*R*^2^ = 0.93874) in dark conditions and [AO7] = 19.52 × exp(−*t*/48.37) + 0.94 (*R*^2^ = 0.98367) under daylight (Appendix A). 

In terms of controls, several different concentrations of **2** have been conducted to check the required concentration that shows significant photocatalytic degradation activity (Appendix A). As decreasing concentration of **2**, the degradation performance decreased but still in significant extents (Figure 7). For all tested concentrations (1 mg **2** dipped into 3, 6, 15, 30 mL AO7_(aq)_), the photocatalytic degradation performances of AO7 are quickly increased to 33–85% within 60-min irradiation and then gradually reach to approximately 59–98% as increasing irradiation time to 1440 min. These results clearly indicated that **2** exhibits good photocatalytic activity to degrade AO7 in the presence of H_2_O_2_ under daylight, even in a concentration as low as 1 mg **2**/30 mL AO7_(aq)_. 

The possible reaction mechanisms are proposed. The preliminary experiments carried out in different conditions have clearly indicated that the degradation of AO7 only taken place when Cu(II) complex (**1** or 2) and H_2_O_2_ were collaborated. In addition, the degradation performance of AO7 by **1/**H_2_O_2_ and **2/**H_2_O_2_ would be significantly improved (acceleration and enhancement) under daylight in compared to that in dark conditions. These observations imply that advanced oxidative processes (AOPs) [22], which is worked by the in situ generated active hydroxyl radical (^•^OH) as oxidizing agent to oxidatively degrade dyes, have been applied to the degradation of AO7 dye and there might be multiple pathways to dominate the AOPs. In general, the water-insoluble Cu(II) complexes can react with H_2_O_2_ to produce ^•^OH radicals via homolytic cleavage or ^•^OH and HO_2_^•^ radicals via Cu redox cycles for degrading AO7 dye both in dark and under daylight [66,67,68]. However, daylight irradiation might result in Cu-based Fenton processes that would cause the generation of photoexcited holes (h^+^) and electrons (e^−^) [66,67,68]. The former might react with OH^−^ to produce ^•^OH whereas the latter might react with H_2_O_2_ to produce ^•^OH or O_2_ to form O_2_^•^^−^ for oxidative degradation of AO7 dye. The formation of more active ^•^OH, HO_2_^•^ and O_2_^•^^−^ radicals accelerates the whole photodegradation process and also enhances the photodegradation performances. 

From the viewpoint of practices, the recycling performance and stability of photocatalysts are of particular importance during photocatalytic reactions [67]. Owing to the high decolorization performances, the recyclability and stability to leaching of **2** has been examined. Recovered powdered samples of **2** were simply washed with deionized water and MeOH several times, which were then used directly for next photocatalytic degradation experiment of AO7 (Appendix A). As observation, the photocatalytic activity of **2** toward AO7 degradation retained. The decolorization performance of **2** after 60-min degradation under daylight after five sequential experiments was slightly reduced by about 16%, giving rise to about 84% degradation efficiency of that in the first time (Figure 8, inset). However, when the irradiation time was extended over 120 min, the decolorization performances were almost unchanged (Figure 8). These results indicate that complex **2** possesses excellent long-term activity and high reusability. Noteworthy, the XRPD patterns of **2** recovered from five sequential photocatalytic degradation experiments of AO7 were in good agreement with that of as-synthesized samples (Figure 4), suggesting high stability in maintaining pristine crystalline phase. On the other hand, inductively coupled plasma optical emission spectrometry (ICP-OES) analyses indicated that there were 0.0620 ± 0.0042 mg/L of Cu(II) ions in the supernatants of AO7 aqueous solutions after 1440-min decolorization by **2** under daylight, which corresponds to 0.128 ± 0.009% Cu(II) ions leached from the solid samples of **2**. The very low ratios of Cu(II) ions in the supernatants suggests that **2** demonstrates high stability to leaching during AO7 degradation in the presence of H_2_O_2_ under daylight. As a result, **2** is a good photocatalytic material toward recyclable AO7 degradation. 

## 4. Conclusions 

In summary, two Cu(II)–salicyaldimine complexes have been successfully synthesized and characterized. Chloro-based complex **1** has the mononuclear structure while nitrato-containing complex **2** adopts a zigzag chain structure. The salicyaldimine ligand in the two complexes is partially deprotonated to be a mono-charged anion and acts as a NNO tris-chelator to bind a Cu(II) center through its pyridine, imino and phenolato groups and leave the neutral carboxyl group free to coordination. Noteworthy, nitrato ligand in **2** suits a bridging-tridentate with a *μ*, *κ*^4^O, O′:O′,O″ coordination mode to bridge two Cu(II) ions, resulted in the formation of extended zigzag chain structure. The results significantly show the critical role of anions in the formation of discrete and polymeric structures of Cu(II)–salicyaldimine complexes. Moreover, **2** would be an excellent photocatalytic material to show remarkable water stability and recyclable photocatalytic degradation activity that is capable of acceleration and enhancement of AO7 decolorization by H_2_O_2_ under daylight.

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
