# Peer review of "Anion-Dominated Copper Salicyaldimine Complexes—Structures, Coordination Mode of Nitrate and Decolorization Properties toward Acid Orange 7 Dye"

_polymers, 2020, doi:10.3390/polym12091910_

Round 1

Reviewer 1 Report

This manuscript reported the synthesis, crystal structures, and photocatalytic decolorization activity of new copper(II) complexes with NNO tridentate Schiff-base ligand. The crystal structures of discrete complex 1 and polymer complex 2 are well determined and characterized. Also, photocatalytic decolorization of AO7 were performed carefully. The reported work has been well done.

I recommend the paper acceptance to Polymers after the following some points were modified.

Comment 1: The decolorization reaction was performed on Complex 2 only. As a reference, the case of using Complex 1 should be discussed or commented.

Comment 2: The reaction mechanism associated with the complex structure should be discussed and proposed.

Author Response

Response to Reviewer 1 Comments

This manuscript reported the synthesis, crystal structures, and photocatalytic decolorization activity of new copper(II) complexes with NNO tridentate Schiff-base ligand. The crystal structures of discrete complex 1 and polymer complex 2 are well determined and characterized. Also, photocatalytic decolorization of AO7 were performed carefully. The reported work has been well done.

I recommend the paper acceptance to Polymers after the following some points were modified.

  • Comment 1: The decolorization reaction was performed on Complex 2only. As a reference, the case of using Complex 1should be discussed or commented.

Response: As suggestion, the decolorization reaction using complex 1 has been carried out. The results showed no decolorization performances in the absence of H2O2 (complex 1 only) both in dark and under daylight, and poor decolorization performances in the presence of H2O2 both in dark and under daylight.  

  • Comment 2: The reaction mechanism associated with the complex structure should be discussed and proposed.

Response: The reaction mechanism has been proposed.

Reviewer 2 Report

The manuscript by Tsai et al. describes the synthesis and structure of two coordination complexes, one of which forms as a one-dimensional coordination polymer. Some physical properties, namely dye decolorization, are also reported.

My first thoughts are that I do not see why this manuscript has been submitted to ‘Polymers’. It does not seem to fit in with the aims and scopes of the journal in any way. Given the emphasis on crystallography of coordination compounds I suggest that Crystals is the appropriate MDPI journal for this work, and my comments below are made based on this recommendation. The manuscript requires major revision before being considered for ‘Crystals’, particularly concerning control experiments, and further evidence of heterogeneity and leaching stability, in the AO7 section.

  • The coordination mode nomenclature that is used is not correct. There is no such thing as eta1 coordination (see IUPAC Red Book). The correct kappa nomenclature should be used.
  • A note for future reference, there is clearly an absorption issue with the polymeric crystal due to the size of the sample used. This can be avoided by using crystals that are not larger than the dimension of the incident X-ray beam (0.3-0.5 mm depending on collimator).
  • The introduction is too short (3 sentences!) and fails to contextualise the main points of the article (especially the activity of complexes towards decolorization).
  • The NMR of the ligand reports too few signals. Not signal is reported for the CH2 group, and there are insufficient protons reported in the aromatic region.
  • Were the reaction heated “to” 80 degrees for 12 hours (implying a ramp) or heated “at” 80 degrees (implying a steady temperature).
  • Microanalysis for compound 2 is slightly outside of the expected normal tolerance. Solvent loss/gain?
  • I am unsure about O-bound hydrogen atoms treatment in the refined models. The text says that they were found in the difference map, but then fixed. If they are found, then they should merely be restrained afterwards (if needed) rather than fixed.
  • Page 4, line 128, should read “protonated and non-coordinated”. Same for line 166, next page.
  • Errors needed (esd) for inter-planar distances (or provide to only 2 decimal places where esd becomes irrelevant). These must also be supplied for other distances where they are missing (e.g. for the “semi-coordination”).
  • Is this mode for nitrate that rare? A quick CSD search finds over 200 examples with these contacts defined as bonds. This number will only increase if the “semi-coordination” is used, especially with the authors claiming very long contacts that I would argue against being any kind of coordination (an electrostatic attraction at best). This should be quantified in the manuscript rather than just claiming it as ‘unusual’.
  • TGA should not feature in the manuscript, a simple line in the experimental section suffices as the graphs merely show the temperature at which decomposition begins.
  • The stability of the crystalline material is not sufficiently demonstrated using PXRD alone if solution phenomena are then investigated. The authors need to show whether there is any leaching of the material into solution (ICP-MS for example) to determine properly whether the properties reported come from a surface effect or from a very dilute concentration of Cu in water. Furthermore, what is the stability in the presence of hydrogen peroxide? There are insufficient checks and controls to confirm the main conclusions. PXRD only confirms that the powder that is present maintains structure, does not in any way demonstrate complete stability to leaching.
  • In terms of controls, what concentration of Cu would be required to show the same response as observed for compound 2. What is the standard proposed mechanism for Cu assisting this process, and is that possible given the coordination environment of the coordination polymer? What activity does compound 1 show for this process, in solution, as a control?

Author Response

Response to Reviewer 2 Comments

The manuscript by Tsai et al. describes the synthesis and structure of two coordination complexes, one of which forms as a one-dimensional coordination polymer. Some physical properties, namely dye decolorization, are also reported.

My first thoughts are that I do not see why this manuscript has been submitted to ‘Polymers’. It does not seem to fit in with the aims and scopes of the journal in any way. Given the emphasis on crystallography of coordination compounds I suggest that Crystals is the appropriate MDPI journal for this work, and my comments below are made based on this recommendation. The manuscript requires major revision before being considered for ‘Crystals’, particularly concerning control experiments, and further evidence of heterogeneity and leaching stability, in the AO7 section.

  • The coordination mode nomenclature that is used is not correct. There is no such thing as eta1 coordination (see IUPAC Red Book). The correct kappa nomenclature should be used.

Response: Thanks for the comment. The coordination mode nomenclature has been described in kappa nomenclature.

  • A note for future reference, there is clearly an absorption issue with the polymeric crystal due to the size of the sample used. This can be avoided by using crystals that are not larger than the dimension of the incident X-ray beam (0.3-0.5 mm depending on collimator).

Response: Thanks for the comment. We will take care in the future.

  • The introduction is too short (3 sentences!) and fails to contextualise the main points of the article (especially the activity of complexes towards decolorization).

Response: The introduction has been re-written. Several new references have been cited.

  • The NMR of the ligand reports too few signals. Not signal is reported for the CH2group, and there are insufficient protons reported in the aromatic region.

Response: Thanks for the comment. The NMR data of the ligand has been correctly reported.

  • Were the reaction heated “to” 80 degrees for 12 hours (implying a ramp) or heated “at” 80 degrees (implying a steady temperature).

Response: Thanks for the comment. The reactions were carried out at 80 degrees for 12 hours. In experimental section, “to” has been changed to “at”. 

  • Microanalysis for compound 2 is slightly outside of the expected normal tolerance. Solvent loss/gain?

Response: Microanalysis for compound 2 has been redone. The present data satisfy the expected normal tolerance.

  • I am unsure about O-bound hydrogen atoms treatment in the refined models. The text says that they were found in the difference map, but then fixed. If they are found, then they should merely be restrained afterwards (if needed) rather than fixed.

Response: As suggestion, the positional parameters of O-bound hydrogen atoms have been refined with the restraint O–H = 0.82(1) Å.

  • Page 4, line 128, should read “protonated and non-coordinated”. Same for line 166, next page.

Response: Thanks for the comment. “remains protonation and un-coordination” has been changed to “is protonated and non-coordinated” in both cases.

  • Errors needed (esd) for inter-planar distances (or provide to only 2 decimal places where esd becomes irrelevant). These must also be supplied for other distances where they are missing (e.g. for the “semi-coordination”).

Response: Thanks for the comment. This manuscript has provided to only 2 decimal places for inter-planar distances, semi-coordination distances, and other distances.

  • Is this mode for nitrate that rare? A quick CSD search finds over 200 examples with these contacts defined as bonds. This number will only increase if the “semi-coordination” is used, especially with the authors claiming very long contacts that I would argue against being any kind of coordination (an electrostatic attraction at best). This should be quantified in the manuscript rather than just claiming it as ‘unusual’.

Response: The word “unusual” has been removed from the revised version of the manuscript.

  • TGA should not feature in the manuscript, a simple line in the experimental section suffices as the graphs merely show the temperature at which decomposition begins.

Response: Thanks for the comment. The decomposition temperatures of 1 and 2 have been provided in the experimental section. However, this manuscript has still kept the section “3.4. Thermogravimetric (TG) Analysis”.

  • The stability of the crystalline material is not sufficiently demonstrated using PXRD alone if solution phenomena are then investigated. The authors need to show whether there is any leaching of the material into solution (ICP-MS for example) to determine properly whether the properties reported come from a surface effect or from a very dilute concentration of Cu in water. Furthermore, what is the stability in the presence of hydrogen peroxide? There are insufficient checks and controls to confirm the main conclusions. PXRD only confirms that the powder that is present maintains structure, does not in any way demonstrate complete stability to leaching.

Response: The ICP-OES analyses of the supernatants of AO7 aqueous solutions after 1440-min decolorization in the presence of 2 and H2O2 indicated 128±0.009% Cu(II) ions leached into the aqueous solutions (0.0620±0.0042 mg/L) from the solid samples of 2, suggesting that 2 demonstrates high stability to leaching during AO7 degradation in the presence of H2O2 under daylight.

  • In terms of controls, what concentration of Cu would be required to show the same response as observed for compound 2. What is the standard proposed mechanism for Cu assisting this process, and is that possible given the coordination environment of the coordination polymer? What activity does compound 1 show for this process, in solution, as a control?

Response: (a) As suggestion, several different concentrations of 2, 1 mg 2 /3, 6, 15, 30 mL AO7(aq), have been conducted to the degradation of AO7 in the presence of H2O2 under daylight. The result showed that 2 exhibits good photocatalytic activity to degrade AO7 in the presence of H2O2 under daylight, even in a concentration as low as 1 mg 2/30 mL AO7(aq),.

(b) The reaction mechanism has been proposed.

(c) The decolorization reaction using Complex 1 has been carried out. The results showed no decolorization performances in the absence of H2O2 both in dark and under daylight, and poor decolorization performances in the presence of H2O2 both in dark and under daylight.   

Reviewer 3 Report

The manuscript is well written. Methods are adequate to the problem and well described.

My main concern is about the H3 atom refinement. 

Since the  hydrogen atoms (H3)  take part in the hydrogen bond, its should be refined ,  not calculated on the base of the known geometry of the molecule.

Please don't use AFIX during refinement or please prove, that in this particular place there is a high residual density and it represents H atom. It could be added to the supplementary material.

Regarding the coordination mode:

line 176-  ....[2.211]. -is it the coordination mode?

line 287- ... μ,η2:η1:η1-tridentate _ is the same coordination mode ? 

Is it the same coordination mode, if yes, how it relates to the description in Scheme 2?

Author Response

Response to Reviewer 3 Comments

The manuscript is well written. Methods are adequate to the problem and well described.

  • My main concern is about the H3 atom refinement.

Since the  hydrogen atoms (H3)  take part in the hydrogen bond, its should be refined ,  not calculated on the base of the known geometry of the molecule.

Please don't use AFIX during refinement or please prove, that in this particular place there is a high residual density and it represents H atom. It could be added to the supplementary material.

Response: As suggestion, the positional parameters of O-bound hydrogen atoms (H3) have been refined with the restraint O–H = 0.82(1) Å.

  • Regarding the coordination mode:

line 176-  ....[2.211]. -is it the coordination mode?

Response: As new reference cited (Russ. Chem. Bull., Int. Ed. 2008, 57, 439–450), the coordination mode of nitrato ligand in 2 has been changed to be “T02” by Morozov notation.

  • line 287- ... μ,η2:η1:η1-tridentate _ is the same coordination mode ?

Is it the same coordination mode, if yes, how it relates to the description in Scheme 2?

Response: As new reference cited (Russ. Chem. Bull., Int. Ed. 2008, 57, 439–450), the coordination mode of nitrato ligand in 2 has been changed to be “T02” by Morozov notation.
